# Sedimentary DNA tracks decadal-centennial changes in fish abundance

Michinobu Kuwae [1,5 ✉], Hiromichi Tamai[1], Hideyuki Doi [2,5], Masayuki K. Sakata[3], Toshifumi Minamoto [3] & Yoshiaki Suzuki [4]

Far too little is known about the long-term dynamics of populations for almost all macro-organisms. Here, we examined the utility of sedimentary DNA techniques to reconstruct the dynamics in the "abundance" of a species, which has not been previously defined. We used fish DNA in marine sediments and examined whether it could be used to track the past dynamics of pelagic fish abundance in marine waters. Quantitative PCR for sedimentary DNA was applied on sediment-core samples collected from anoxic bottom sediments in Beppu Bay, Japan. The DNA of three dominant fish species (anchovy, sardine, and jack mackerel) were quantified in sediment sequences spanning the last 300 years. Temporal changes in fish DNA concentrations are consistent with those of landings in Japan for all three species and with those of sardine fish scale concentrations. Thus, sedimentary DNA could be used to track decadal-centennial dynamics of fish abundance in marine waters.

[1] Center for Marine Environmental Studies, Ehime University, Matsuyama 790-8577, Japan. [2] Graduate School of Simulation Studies, University of Hyogo, Kobe 650-0047, Japan. [3] Graduate School of Human Development and Environment, Kobe University, Kobe 657-8501, Japan. [4] Research Institute of Geology and Geoinformation, Geological Survey of Japan, National Institute of Advanced Industrial Science and Technology, Tsukuba 305-8567, Japan. [5]These authors contributed equally: Michinobu Kuwae, Hideyuki Doi. ✉email: mkuwae@sci.ehime-u.ac.jp

Approaches using environmental DNA (eDNA) are providing valuable insights on ancient environments, and are proving useful for monitoring contemporary biodiversity in terrestrial and aquatic ecosystems[1,2]. The sedimentary DNA (sedDNA) of organisms is a type of eDNA that is preserved for long periods[3,4], and could be used as successive proxy data for aquatic organisms at a place lacking lengthy monitoring observations. At present, only a very few studies have been conducted on the sedDNA of macro-organisms. These studies investigated a variety of subjects; i.e., major changes in species composition[4,5], extinct biota[4,5], livestock farming history associated with the Anthropocene[3], and the identification of a taxon as a native or alien fish species in lake ecosystems[6,7]. These studies demonstrated that sedDNA is a potentially powerful tool that could be used to detect the "presence/absence" of a taxon and reconstruct past species compositions for macro-organisms. However, it remains unclear whether sedDNA is also powerful to reconstruct the "abundance", which is defined as both the number of individuals and biomass here, of a taxon during the instrumental and pre-instrumental era.

Long-term variability in abundance of a macro-organism could provide fundamental information to evaluate its evolution, responses to climate changes, and human impacts, and to set managements and preservation strategies. For marine fish species, long-term fish abundance records based on landing data have been used by many studies on biological and physical fields of research[8,9], fisheries management[10], and various analyses associated with issues on economic, food, and ecosystem security[11,12]. Historical catch records of marine fishes reveal synchronous large variability over multi-decadal timescales virtually everywhere where available[9,13–15]. This basin-wide synchronous variation might be caused by climate-induced ocean regime shifts[8,9,16], reflecting internal climate variability, such as the Pacific decadal oscillation[9]. However, many of the periods capture only one major "cycle" of decadal variability, and are confounded by the potential effects of fishing and other anthropogenic factors. Thus, records that extend beyond the currently available historical period are needed to define variability in fish species abundance accurately, and their responses to long-term climate change. This information could ultimately be used to inform fishery management, fishery resource utilization, and risk assessment for species extinction in the coming decades and century.

Sedimentary records of abundance of a macro-organism species spanning more than a century are rare in aquatic systems. Such records tend to only be reconstructed for fish species[17–22]. Yet, just 8% of fish species in stocks annually evaluated by Food and Agriculture Organization of the United Nations (72 taxa[23]) have been addressed by previous studies. There remains a distinct lack of information on long-term changes in abundance for many marine fish species and other macro-organisms in aquatic systems. Thus, sedDNA could prove effective in overcoming this issue.

Recent eDNA studies have demonstrated that the abundance of fish species could be estimated by detecting and quantifying eDNA[24,25]. DNA copies in water might be positively correlated with abundance, thus, eDNA might be an effective tool for estimating abundance in aquatic systems[24,26]. Much denser concentrations of fish DNA are detected in bulk-sediment compared to overlying waters, with fish DNA concentrations in water and sediment being correlated[27]. DNA analyses of lake sediment sequences showed that the DNA of fish species was present in layers that were deposited more than 1000 years ago[6,28]. Thus, fish DNA might be preserved in sediment sequences for a long time. Collectively, sedimentary sequences might preserve records of temporal changes in fish DNA concentrations, which are caused by temporal changes in the abundance of fish, and the resultant sedimentation rates of their DNA. However, studies are required to determine whether temporal variation in DNA concentration can be used to track past dynamics in fish abundance in aquatic systems.

Sediments in Beppu Bay, Japan (Supplementary Fig. 1), provide suitable samples for detecting fish sedDNA, and are ideal for evaluating the potential of using sedDNA to reconstruct fish abundance. Because anoxic conditions in the bottom waters of the innermost part of Beppu Bay persisted during the late Holocene[29], these conditions reduce nuclease degradation, favoring the long-term preservation of DNA[30]. A previous study suggested that the abundance of anchovy and sardine fish scales in the sediments reflects past fish abundance in the overlying water[17]. Thus, fish scales could be used to test whether DNA concentrations reflect fish abundance by comparing these proxies over long-time frames. Furthermore, landing data were recorded for >50 years in nearby areas and >100 years for Japan, with good and poor sardine catch histories being documented for >300 years in an area adjacent to a sardine spawning ground (Bungo Channel). This information could also be used to test whether DNA concentrations reflect fish abundance by comparing sedDNA with landing data.

Here, we explore the DNA of three major fish species (Japanese anchovy, *Engraulis japonicus*, Japanese sardine, *S. melanostictus*, and jack mackerel, *Tranchurus japonicus*) in sediment core samples from Beppu Bay spanning 300 years by applying quantitative polymerase chain reaction (qPCR) methods (Experiment (a) in Fig. 1). We used this approach to test the utility of sedDNA to reconstruct past fish abundance by comparing fish sedDNA concentrations with fish scale concentrations and landing data in the published literature. Furthermore, we examined the sources of DNA in bulk sediments using DNA extracted from potential sources, including fish scales and bones in the sediment, various particle size fractions in the sediment (Experiment (b)), and pore water (Experiment (c)). Potential environmental factors driving sedDNA concentrations were also considered, including enzymatic inhibitors during PCR, dilution of soil materials, and aerobic microbial degradation rates of DNA (Experiment (d)). Our results demonstrate that sedDNA records past decadal–centennial changes in given fish species abundance in the water and may be an independent proxy of fish abundance, in addition to fish scales which were used in previous studies[17–20]. This finding highlights the utility of sedDNA to reconstruct macro-organism species abundance.

## Results

**DNA concentration in core sediments.** qPCR analyses for each core (Experiment (a) in Fig. 1) showed that the mean DNA copies for anchovy ranged from $233 \pm 215$ copies $g^{-1}$ dry sediment (hereafter, copies $g^{-1}$, mean $\pm 1$ SD) to $3075 \pm 781$ copies $g^{-1}$ (Supplementary Table 1). For all data, anchovy had $1067 \pm 968$ mean DNA copies $g^{-1}$. Anchovy DNA copies were detected in all samples, except three (13 cm depth in BG17-1, 41 and 45 cm in BMC18-6). For most samples, DNA was detected in more than three out of four replicates. In each core, the mean DNA copies for sardines ranged from $1.7 \pm 4.0$ to $12.0 \pm 29.2$ copies $g^{-1}$. For all data, sardine had $5.1 \pm 3.8$ mean DNA copies $g^{-1}$. This concentration was 0.5% that of anchovy. Sardine DNA copies were not detected in many of the samples from each core. One or two replicates were detected in most samples, while a few samples had more than three out of four replicates for sardines. Jack mackerel DNA copies were also not detected for many samples from each core. Mean DNA copies for each core ranged from 0 to $55.9 \pm 127.2$ copies $g^{-1}$. For all data, Jack mackerel had $14.8 \pm 19.1$ mean DNA copies $g^{-1}$. This concentration was 1.4% that of anchovy.

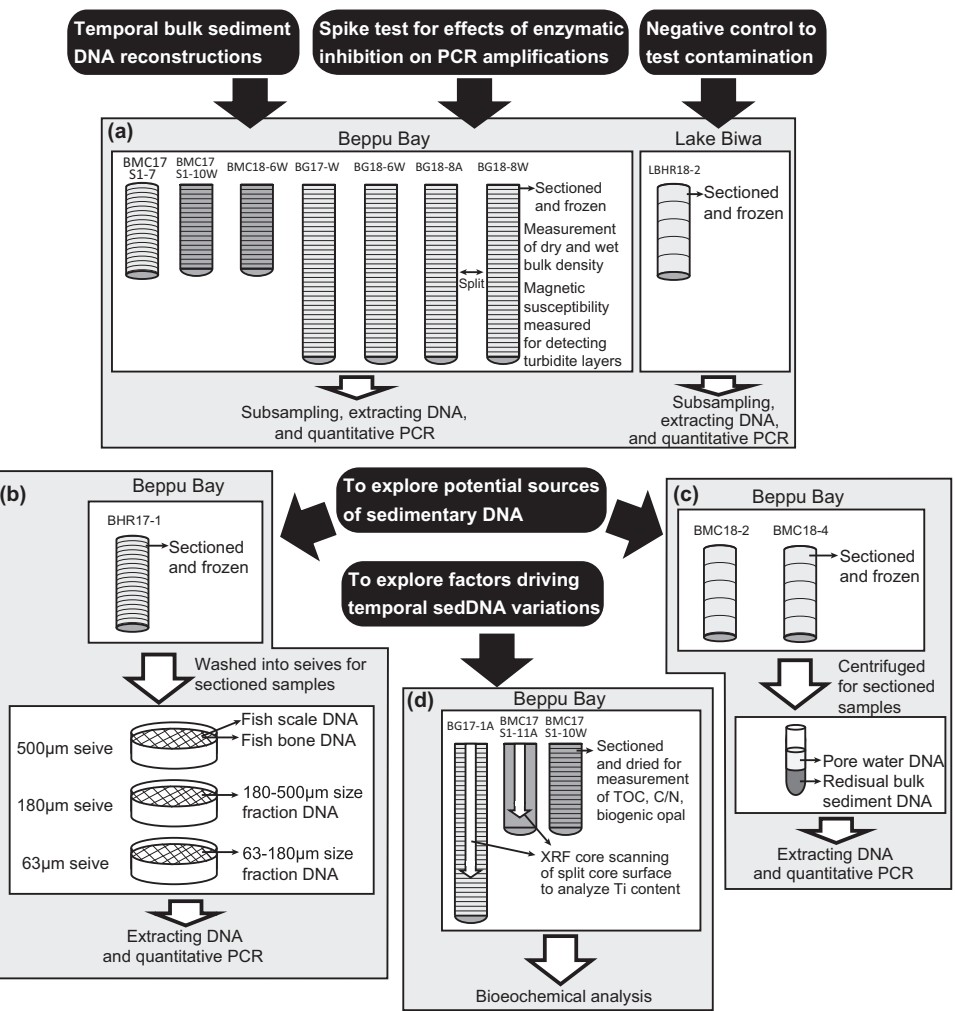

**Fig. 1 Experimental design for this study.** Experiment (a) reconstruction of temporal variation in bulk-sediment DNA concentrations; (b) measurement of fish scale and bone DNA and DNA in coarse and fine particle size fractions; (c) measurement of pore water DNA; and (d) measurement of environmental indices related to potential factors driving temporal variations in sedimentary DNA.

Few samples had more than three out of four replicates, with most samples having one or two replicates.

For the negative control for sectioning, subsampling, DNA extraction, and PCR processes (Experiment (a) in Fig. 1), DNA of the three marine species was not detected in any of the core sediment samples from Lake Biwa (LBHR18-1) (Supplementary Table 1). DNA was not detected on the PCR blanks either (Supplementary Table 1). Thus, contamination was not an issue during sampling, extraction, purification, and the PCR processes. Also, through the direct sequencing of PCR amplicons by the qPCR assay of Japanese sardine, we confirmed that only the DNA was amplified.

We performed spike test to evaluate the effect of PCR inhibition (Experiment (a) in Fig. 1). All ΔCt values were less than three (ΔCt: −2.4–2.9) (Supplementary Table 2), providing no evidence of inhibition[31].

**Down-core changes in DNA concentration**. Down-core changes in DNA concentration for anchovy showed different patterns between each core (Supplementary Figs. 2 and 3) (Experiment (a) in Supplementary Fig. 1). For 50-cm-long core samples, peaks in DNA occurred at around 5 and 20 cm in BMC18-6, but occurred at around 0 cm in BMC17 S1-7 (Supplementary Fig. 2). There was no noticeable peak in BMC17 S1-10 (Supplementary Fig. 2). For 110-cm-long core samples, there was no consistent vertical

pattern, except for the uppermost layers, with the highest values being detected for BG18-6W and BG18-8A (Supplementary Fig. 3). For sardine and jack mackerel, there were no consistent vertical patterns in DNA for the short cores. In contrast, the 1.1-m-long cores had peaks centered at around 16 and 57 cm deep for sardine and at around 20 cm for jack mackerel. Comparison between short and long cores (Supplementary Figs. 2 and 3) showed that anchovy and jack mackerel had the highest DNA concentrations in the uppermost layers in the short cores. The long cores did not show a similar trend, due to loss of surface layers (approximately 20 cm) during core collection. In contrast, the highest DNA concentrations for sardine occurred at 57–58 cm depth in the long cores, not in the uppermost layers of the short cores.

There was no clear evidence that DNA concentrations were higher in core samples that were instantly frozen after core collection (core BMC17 S1-7) compared to samples that were frozen 6 days or 4 weeks after core collection (Supplementary Figs. 2 and 3) (Experiment (a) in Fig. 1). Thus, chilled storage for 4 weeks only caused minor degradation of DNA in core samples.

**Temporal changes in DNA concentration**. General additive models (GAMs) showed that the decadal–centennial dynamics of the inter-core, seven-year averaged, and sedDNA concentrations for the last 300 years significantly varied non-linearly (Japanese

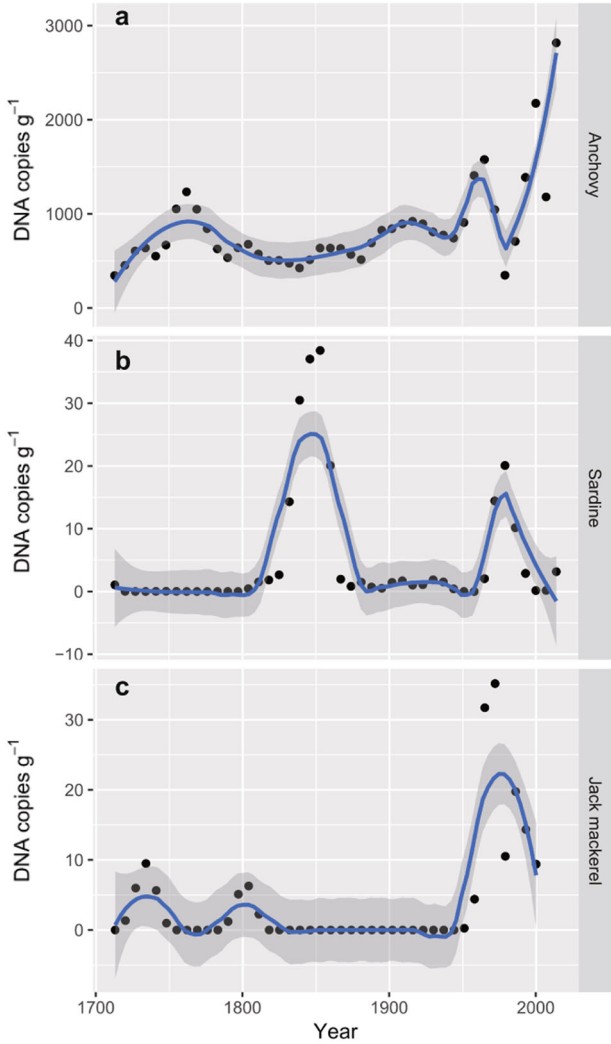

**Fig. 2 The results of general additive models (GAM) from inter-core, seven-year averaged sedDNA concentrations. a** *Engraulis japonicus* (Japanese anchovy); **b** *Sardinops melanostictus* (Japanese sardine); **c** *Tranchurus japonicus* (jack mackerel). Blue line denotes a regression line of GAM with the 95% confidence interval (gray zone).

anchovy, $s = 7.22$, $P = 2.96 \times 10^{-7}$; Japanese sardine, $s = 12.61$, $P = 1.10 \times 10^{-4}$; jack mackerel, $s = 8.831$, $P = 2.84 \times 10^{-9}$, Fig. 2). DNA concentrations for Japanese anchovy were high after 2010 CE (BMC18-6, BMC17 S1-7, and BMC17 S1-10) (Fig. 3). While there was no consistent pattern in the time series of the cores before 2000 CE, one or two of the time series showed high values around the 1960s CE and the 2000 CE. These periods with high values showed large scatters between the cores, indicating spatial heterogeneity in DNA deposition.

These high values were not obtained in the anchovy fish scale concentrations during the same periods[17] (Supplementary Fig. 4). There was also no significant relationship in the Type II regression model for the inter-core, seven-year averaged, concentrations in DNA (Fig. 4a) with those of fish scale concentrations (Fig. 4c) for the 300 years ($R^2 = 0.0157$, $P = 0.429$, $n = 42$, Fig. 5a, also see Supplementary Fig. 5 for the log-transformed model). In contrast, the two DNA peaks in the 1960s and 2000 were temporally consistent with those of the catch record in Japan (Statistics of Agriculture, Forestry and Fisheries, with the landing data being updated from previous studies[32,33]) (Fig. 4b). This result was supported by a Type II regression

model, with a significant correlation existing between inter-core, seven-year averaged concentrations in DNA (Fig. 4a) and seven-year averaged catches in Japan (Fig. 4b) ($R^2 = 0.255$, $P = 0.0459$, $n = 16$, Fig. 6a, also see Supplementary Fig. 6a for the log-transformed model). Anchovy sedDNA and landings in Japan before 1990 showed a positive-phase relationship with the Bungo Channel (Supplementary Fig. 7, see Supplementary Fig. 1a for the location), but a negative-phase relationship with the central Seto Inland Sea (Supplementary Fig. 7c). A decadal peak around 2000, as shown by the sedDNA and landings in Japan, was not obtained in the landings from the Bungo Channel and Beppu Bay, Iyo-nada, and Suo-nada (Supplementary Fig 7a, b, see Supplementary Fig. 1 for the locations and see Supplementary Discussion for the reasons). An abnormally high value in 2014–2017 was not found in the landing records (Supplementary Fig 7a, b). This inconsistency indicates the presence of enriched DNA in the surface layer that is susceptible to rapid decomposition due to early diagenesis in a few years.

DNA concentrations for Japanese sardine showed large scatters of contemporary values between the cores (Fig. 3b), indicating spatial heterogeneity in DNA depositions. The concentrations were high (>20 copies $g^{-1}$) during the 1840s to 1850s and 1970s to 1980s for the time series of the three cores, and were low (<20 copies $g^{-1}$) during the other periods (Fig. 3b). GAMs showed that the peaks during the 1840s to 1850s were comparable to, or higher than, those during the 1970s to 1980s (Fig. 2). These two peaks were consistent in time with the high values (>0.25 $g^{-1}$) recorded for sardine fish scale concentrations (Supplementary Fig. 4b, f); however, there was no noticeable peak during the 1920s and 1930s (Figs. 3b and 4b), despite a peak occurring in the fish scale record (Fig. 4f and Supplementary Fig. 4b). The peak in DNA during the1970s to 1980s corresponded to a distinct peak in sardine catches during the twentieth century (Fig. 4e). The DNA peak in the 1840s to 1850s was consistent with good catch periods recorded by historical documents in and around the Bungo Channel[34,35] (Fig. 3b). Sardine DNA was detected during ~1700 CE (14 copies $g^{-1}$), which was consistent with a good catch period recorded in the historical documents (Fig. 3b), and a minor peak in the fish scale record (Fig. 4f and Supplementary Fig. 4b). Type II regression for sardine showed a significant correlation between inter-core, seven-year averaged concentrations of DNA (Fig. 4d) and fish scale concentrations (Fig. 4f) for the last 300 years ($R^2 = 0.436$, $P = 1.93 \times 10^{-6}$, $n = 42$, Fig. 5b, also see Supplementary Fig. 5b for the log-transformed model). It also showed a significant correlation between inter-core, seven-year averaged concentrations of DNA and seven-year averaged catches in Japan ($R^2 = 0.269$, $P = 0.0395$, $n = 16$, Fig. 6b, also see Supplementary Fig. 6b for the log-transformed model). sedDNA and landings in Japan showed a positive-phase relationship with Bungo Channel. However, a clear relationship was not detected with the landings in Beppu Bay, Iyo-nada, and Suo-nada or the central Seto Inland Sea (Supplementary Fig. 8, see Supplementary Discussion for the reasons). There was a negative phase relationship between sardine and anchovy in sedDNA after the 1950 CE (Fig. 2a, b).

Jack mackerel DNA concentrations for each core (Fig. 3c) were high, exceeding 50 copies $g^{-1}$ around 1970 and 1990 for the two core time series, and exceeding 100 copies $g^{-1}$ after 2005, with low values (<20 copies $g^{-1}$) being obtained during the other periods. Before 1965, DNA values were low, with less than 50 copies $g^{-1}$. There was no significant correlation between inter-core, seven-year averaged concentrations of jack mackerel DNA (Fig. 4g) and landings in Japan (Fig. 4h) ($R^2 = 0.0786$, $P = 0.501$, $n = 8$, Fig. 6c, also see Supplementary Fig. 6 for the log-transformed model). This non-correlation might be due to the small sample size (short record of jack mackerel landing).

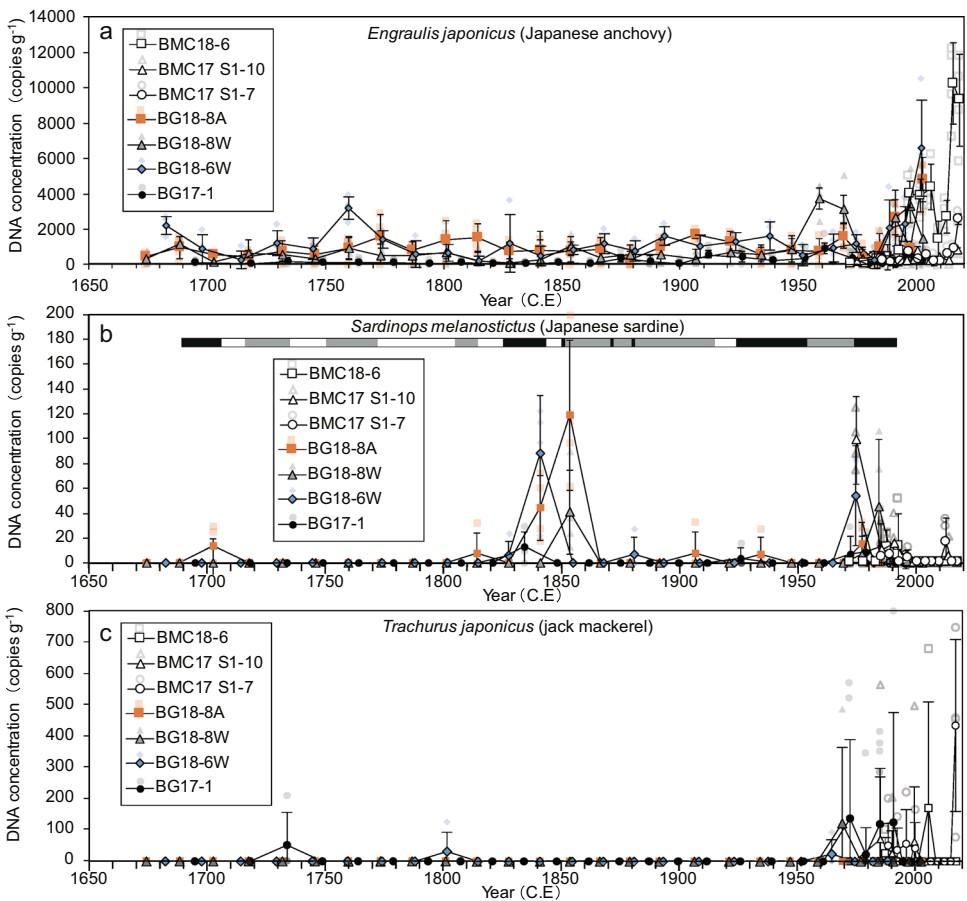

**Fig. 3 Temporal changes in mean DNA concentrations for all cores. a** Japanese anchovy; **b** Japanese sardine; **c** jack mackerel. Error bar of each data point denotes 1 SD ($n = 4$ or 8). The horizontal bar in panel **b** represents historical good (solid) and poor (gray) catch periods (open: no data). Translucent colored plots denote each data point in qPCR replicates.

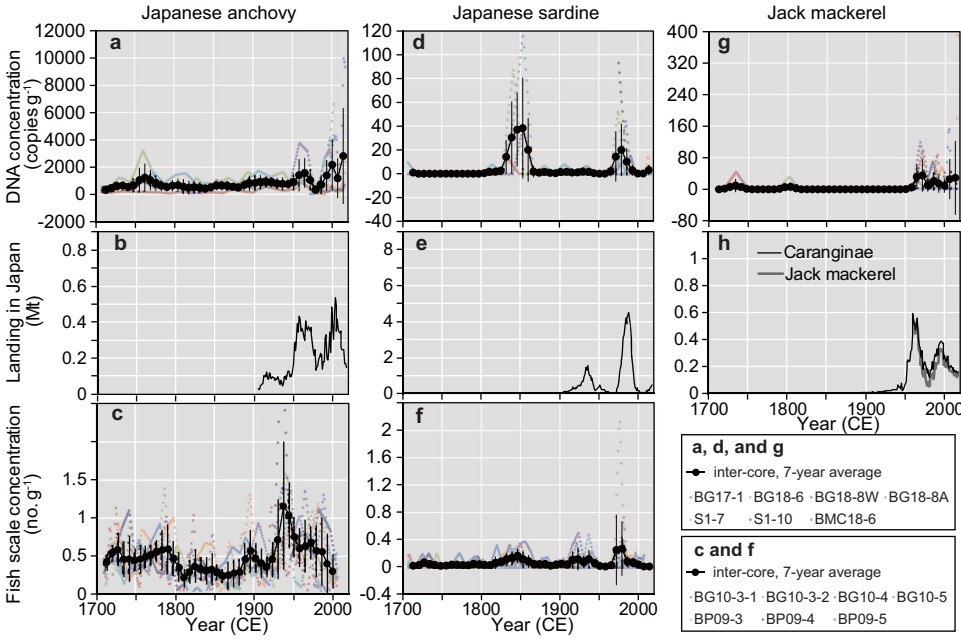

**Fig. 4 Comparison between temporal changes in sedDNA concentrations, landings, and fish scales. a**, **d**, and **g**: inter-core, seven-year averaged concentrations of DNA for anchovy (left), sardine (middle), and jack mackerel (right). **b**, **e**, and **h**: total landings in Japan. **c** and **f**: fish scales. Of note, the landings of Caranginae (jack mackerel plus amberstripe scad, *Decapterus muroadsi*) consist mostly of those of jack mackerel. Error bar of each data point denotes 1 SD. Translucent colored plots denote annual data points for each core.

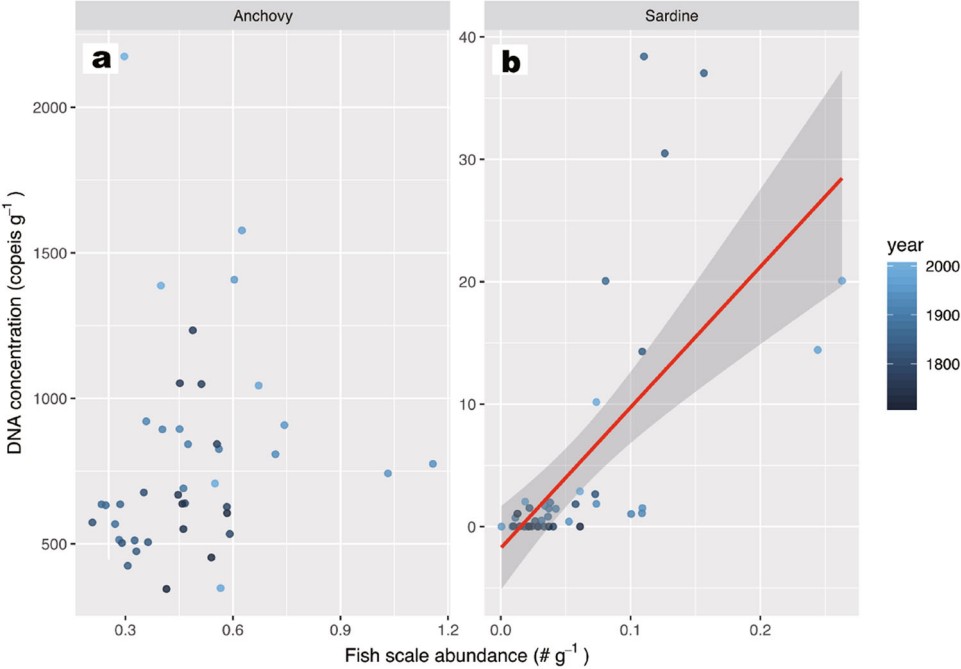

**Fig. 5 Relationships between sedDNA and fish scale concentrations. a**: Japanese anchovy; and **b**: Japanese sardine. Inter-core, 7-year average data were used for the models. Red line denotes a regression line of Gaussian Type II regression model with the 95% confidence interval (gray zone).

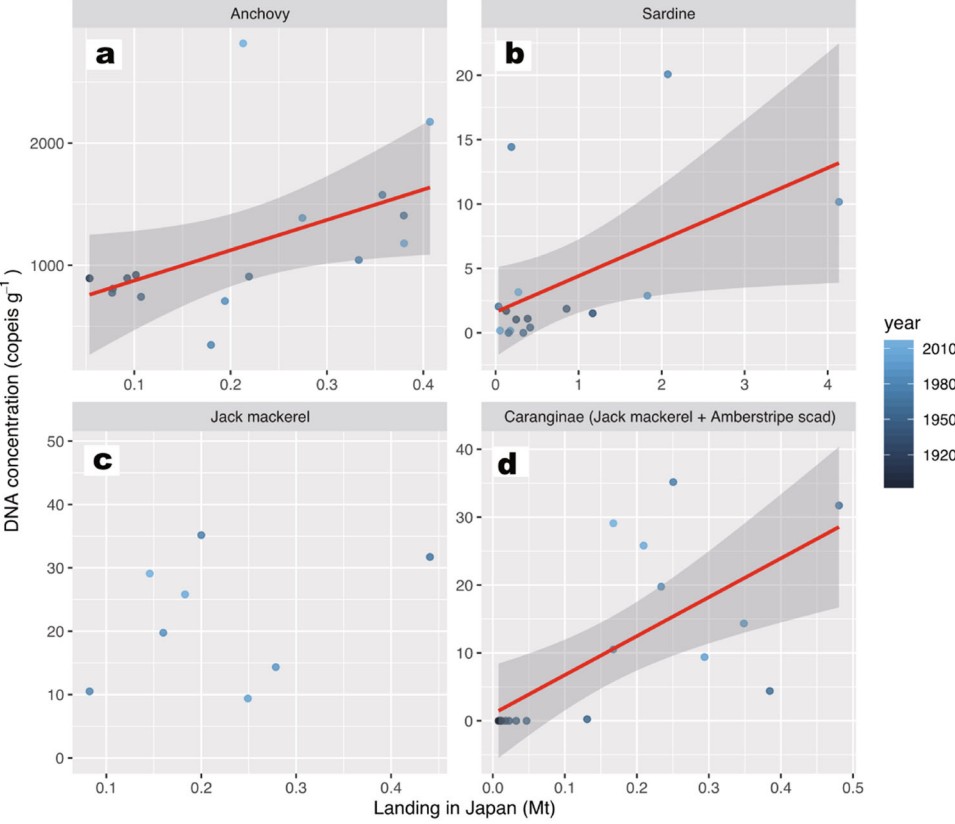

**Fig. 6 Relationships between sedDNA concentrations and the total landings in Japan. a**: Japanese anchovy; **b**: Japanese sardine; **c**: jack mackerel; and **d**: Caranginae (jack mackerel and amberstripe scad). Inter-core, 7-year average data for eDNA and 7-year average data for landing were used in the models. Red line denotes a regression line of Gaussian Type II regression model with the 95% confidence interval (gray zone).

**Table 1 DNA copies for each species for pore water and pore water-free sediment samples.**

| Sample ID | Core depth | | Pore water (copies ml$^{-1}$) | SD | Detection/total replicates | Pore water-free sediment mean (copies g$^{-1}$ dry) | SD | Detection/ total replicates |
|---|---|---|---|---|---|---|---|---|
| | Top (cm) | Bottom (cm) | | | | | | |
| *Japanese anchovy* | | | | | | | | |
| BMC18-2 | 0 | 5 | ND | — | 0/4 | 6422.6 | 1140.7 | 4/4 |
| BMC18-2 | 20 | 25 | ND | — | 0/4 | 2597.9 | 684.2 | 4/4 |
| BMC18-2 | 40 | 45 | ND | — | 0/4 | 367.1 | 290.0 | 3/4 |
| BMC18-4 | 0 | 5 | ND | — | 0/4 | 3469.7 | 1915.8 | 4/4 |
| BMC18-4 | 20 | 25 | ND | — | 0/4 | 2904.3 | 1093.6 | 4/4 |
| BMC18-4 | 30 | 35 | ND | — | 0/4 | 466.7 | 281.1 | 4/4 |
| *Japanese sardine* | | | | | | | | |
| BMC18-2 | 0 | 5 | ND | — | 0/4 | 0.0 | 0.0 | 0/4 |
| BMC18-2 | 20 | 25 | ND | — | 0/4 | 0.0 | 0.0 | 0/4 |
| BMC18-2 | 40 | 45 | ND | — | 0/4 | 13.5 | 15.5 | 2/4 |
| BMC18-4 | 0 | 5 | ND | — | 0/4 | 0.0 | 0.0 | 0/4 |
| BMC18-4 | 20 | 25 | ND | — | 0/4 | 283.5 | 567.0 | 1/4 |
| BMC18-4 | 30 | 35 | ND | — | 0/4 | 12.4 | 24.8 | 0/4 |
| *Jack mackerel* | | | | | | | | |
| BMC18-2 | 0 | 5 | ND | — | 0/4 | 0.0 | 0.0 | 0/4 |
| BMC18-2 | 20 | 25 | ND | — | 0/4 | 0.0 | 0.0 | 0/4 |
| BMC18-2 | 40 | 45 | ND | — | 0/4 | 0.0 | 0.0 | 0/4 |
| BMC18-4 | 0 | 5 | ND | — | 0/4 | 0.0 | 0.0 | 0/4 |
| BMC18-4 | 20 | 25 | ND | — | 0/4 | 84.6 | 169.2 | 1/4 |
| BMC18-4 | 30 | 35 | ND | — | 0/4 | 0.0 | 0.0 | 1/4 |

Therefore, we further compared temporal variation in jack mackerel DNA with longer-available time series of *Caranginae* in Japan, which did not discriminate between landings of Jack mackerel and Amberstripe scad before 1960. However, variation in the annual catch of *Caranginae* might reflect that of jack mackerel, because jack mackerel catches after 1960 accounted for $79 \pm 14\%$ of *Caranginae* landings (sum of jack mackerel and amberstripe scad). There was a significant correlation between inter-core, 7-year averaged concentrations of jack mackerel DNA (Fig. 4g) and 7-year averaged catches of *Caranginae* in Japan (Fig. 4h) ($R^2 = 0.453$, $P = 0.00221$, $n = 18$, Fig. 6d, also see Supplementary Fig. 6d for the log-transformed model). Jack mackerel sedDNA and landings in Japan showed a positive-phase relationship with landings from the Bungo Channel, and a weak positive-phase relationship with landings from Beppu Bay, Iyo-nada, and Suo-nada (Supplementary Fig. 9). The 2017 peak was not found in the landing records (Supplementary Fig. 9). This inconsistency indicates the presence of enriched DNA in the surface layer that is susceptible to rapid decomposition due to early diagenesis in a few years.

**Relationship between anchovy DNA and environmental factors.** Anchovy DNA copies showed a weak significant negative correlation with titanium (Ti) contents (a soil mineral tracer[36]) ($r = -0.41$, $P = 0.011$), and no significant correlation with total organic carbon contents (TOC, a mixture of terrestrial and marine-derived organic carbon), molar carbon and nitrogen ratios (C/N, an index of supply rate of terrestrial organic matters[37]), biogenic opal (mainly originated from diatom valves), and sedimentation rates (Supplementary Table 3). In the upper laminated layers (Supplementary Figs. 11 and 12), anchovy DNA showed no correlation with any of the environmental indices ($r = -0.36$–$0.08$, $P > 0.05$, Supplementary Table 3). In the lower massive layers (Supplementary Figs. 11 and 12), anchovy DNA showed a significant positive correlation with TOC ($r = -0.50$, $P = 0.029$) and biogenic opal ($r = 0.47$, $P = 0.044$), and a negative correlation with C/N ($r = -0.48$, $P = 0.040$). It showed no correlation with Ti and sedimentation rate.

**Source materials of DNA in marine sediments.** The DNA in the pore water of each sample (Experiment (c) in Fig. 1) was not detected by qPCR assays for any of the species (Table 1). In contrast, anchovy DNA was detected in the residual bulk sediments of all samples (range: 367–6423 copies g$^{-1}$, mean: $2704 \pm 2233$ copies g$^{-1}$), while sardine DNA was detected in two samples (range: 12.4–283.5 copies g$^{-1}$, mean: $51.6 \pm 113$ copies g$^{-1}$) and jack mackerel was detected in one sample (84.6 copies g$^{-1}$) (Table 1, Experiment (c) in Fig. 1). DNA was only detected in the fish scales of anchovy ($1.5 \pm 4.2$ copies scale$^{-1}$) (Table 2) (Experiment (b) in Fig. 1). DNA from bones was not detected in any of the species (Table 2) (Experiment (b) in Fig. 1). DNA was detected in the 63–180 μm size fractions of one sample for sardine ($0.9 \pm 1.7$ copies g$^{-1}$ dry sediment before sieved) and jack mackerel ($0.3 \pm 0.6$ copies g$^{-1}$ dry sediment before sieved), but was not detected for anchovy (Table 2). DNA was not detected in the 180–500 μm fractions for any of the species (Table 2).

**Discussion**
By using the sedDNA approach, we detected anchovy, sardine, and jack mackerel DNA from sediment layers dating back 300 years, confirming that fish DNA can be detected from marine sedimentary sequences. Our finding supports the long-term preservation potential of macro-organism DNA in the sediments of marine systems, as well as freshwater systems as previously reported[3,6,7], even though aquatic and bottom environments are completely different in marine and freshwater systems (e.g., salinity, pH, light, and tidal effects).

Our data also identified decadal–centennial variation in the sedDNA concentrations of fish over the last 300 years. Sardine sedDNA values were higher around 1850 CE compared to the peak at around 1980. Thus, time-dependent diagenetic degradation of fish sedDNA seems to be minor at centennial timescales, although sedDNA diagenetic degradation and underestimation of the older DNA deposition rates cannot completely be excluded. These results imply that sedDNA signatures track temporal variation in fish abundance.

Several factors might drive sedDNA concentrations, other than fish abundance. These include contamination of enzymatic

**Table 2 DNA copies for each species for fish scales, bones, and fine (63–180 μm) and coarse (180–500 μm) particle size fraction of sediment samples.**

| Material<br>Species<br>Core sample ID | Core depth | | Mean DNA copies | SD | Detection/total replicates |
|---|---|---|---|---|---|
| | Top (cm) | Bottom (cm) | | | |
| **Fish scale** | | | (copies per scale) | | |
| *Japanese anchovy* | | | | | |
| BHR17-1 | 0 | 36 | 1.5 | 4.2 | 1/8 |
| *Japanese sardine* | | | | | |
| BHR17-1 | 0 | 36 | ND | — | 0/4 |
| *Jack mackerel* | | | | | |
| BHR17-1 | 0 | 36 | ND | — | 0/4 |
| **Bone** | | | (copies per bone) | | |
| *Japanese anchovy* | | | | | |
| Mixed core sample | | | ND | — | 0/4 |
| *Japanese sardine* | | | | | |
| Mixed core sample | | | ND | — | 0/4 |
| *Jack mackerel* | | | | | |
| Mixed core sample | | | ND | — | 0/4 |
| **63–180 μm size fraction** | | | (copies g$^{-1}$ dry sediment before sieved) | | |
| *Japanese anchovy* | | | | | |
| BHR17-1 | 20 | 21 | ND | — | 0/4 |
| BHR17-1 | 25 | 26 | ND | — | 0/4 |
| BHR17-1 | 27 | 28 | ND | — | 0/4 |
| *Japanese sardine* | | | | | |
| BHR17-1 | 20 | 21 | ND | — | 0/4 |
| BHR17-1 | 25 | 26 | 0.9 | 1.7 | 1/4 |
| BHR17-1 | 27 | 28 | ND | — | 0/4 |
| *Jack mackerel* | | | | | |
| BHR17-1 | 20 | 21 | ND | — | 0/4 |
| BHR17-1 | 25 | 26 | ND | — | 0/4 |
| BHR17-1 | 27 | 28 | 0.3 | 0.6 | 1/4 |
| **180–500 μm size fraction** | (copies g$^{-1}$ dry sediment before sieved) | | | | |
| *Japanese anchovy* | | | | | |
| BHR17-1 | 20 | 21 | ND | — | 0/4 |
| BHR17-1 | 25 | 26 | ND | — | 0/4 |
| BHR17-1 | 25 | 28 | ND | — | 0/4 |
| *Japanese sardine* | | | | | |
| BHR17-1 | 20 | 21 | ND | — | 0/4 |
| BHR17-1 | 25 | 26 | ND | — | 0/4 |
| BHR17-1 | 27 | 28 | ND | — | 0/4 |
| *Jack mackerel* | | | | | |
| BHR17-1 | 20 | 21 | ND | — | 0/4 |
| BHR17-1 | 25 | 26 | ND | — | 0/4 |
| BHR17-1 | 27 | 28 | ND | — | 0/4 |

inhibitors during PCR, dilution of soil materials, sediment burial rates, and oxygenation in the sediment–water interface (resulting in aerobic microbial degradation rates of DNA during early diagenesis), co-sinking with phytoplankton debris and/or their aggregations, and scavenging by flood materials from rivers. Since we confirmed no clear relationship between anchovy DNA concentrations and proxies of these factors (for details, see Supplementary Discussion), decadal–centennial variation was not explained by temporal changes in these factors.

We showed significant relationships between sedDNA and landings in Japan for all three species. The positive relationships were considered reasonable for several reasons. Beppu Bay is located near waters off south Japan that are the main spawning areas of the Pacific stock for Japanese sardine, which is the largest stock around Japan. The growing areas of Japanese sardine shrank and extended at decadal timescales, whereas the main spawning areas remained stable in the waters off south Japan. Therefore, waters close to spawning areas like Beppu Bay

are most appropriate for producing a record of regional sardine abundance in the western North Pacific[17]. In fact, annual recruitment of sardine in the Seto Inland Sea (including Beppu Bay) showed a good correlation with that in the main spawning areas[38]. Therefore, sedDNA in a few grams of sediment sample could effectively track DNA sedimentation rates that are proportional to sardine abundance in Beppu Bay, and that probably link to the landings of Pacific stock in Japan. The same reasoning could be applied for Japanese anchovy and jack mackerel, because the spawning areas of the Pacific stocks of Japanese anchovy and jack mackerel are also located in waters off south Japan.

Our results demonstrated alternations in peaks between anchovy and sardine, which is a well-known marine ecological phenomenon, termed "regime shifts," documented in the Pacific[14,39,40]. Thus, we confirmed that sedDNA could be used to track decadal-scale changes in fish abundance in water overlying bottom sediment sequences.

The relationship between sedDNA and fish scales over the last 300 years differed between species (Fig. 5). Sardine sedDNA captured two peaks (Fig. 4d) recorded from fish scales and historical documents. Thus, sedDNA could be used to infer fish abundance for this species during the pre-instrumental era. In contrast to sardine, no clear relationship between sedDNA and fish scales was found for anchovy (Fig. 5a). Thus, sedDNA and fish scales might have different signatures with respect to the population dynamics of anchovy.

Alternative explanation of the documented inconsistency between sedDNA and fish scale concentrations might result from large scatters of contemporary inter-core data for both sedDNA and fish scales. Spatial variation in fish eDNA concentrations in water was observed in Maizuru Bay[25], indicating heterogeneous distributions of fish in this area. Spatial heterogeneity in fish abundance might cause spatial variation in DNA and fish scale sedimentation and concentrations in the surface sediments. Therefore, the lack of consistency between temporal variation in sedDNA and fish scales could be resolved by the corresponding peaks being potentially detected by several additional reconstructions. To acquire real signatures of temporal variation in fish abundance, it is necessary to integrate inter-core averaged data of three time series or more for both sedDNA and fish scales.

Anchovies in the Bungo Channel originate from the early-spring Pacific-generated population, which is one of the subpopulations of the Seto Inland Sea stock. This population spawns and grows in and off the Bungo Channel, and migrates to the western Seto Inland Sea[41] (see Fig. 10 in ref. [17], for the migration routes of the inland sea subpopulations); however, this subpopulation cannot be discriminated from the Pacific stock, which is the largest population of Japanese anchovy, because the two groups use the same spawning area in the same season. A positive-phase relationship between anchovy sedDNA and anchovy landings before 1980 CE in the Bungo Channel and Japan indicated that anchovy sedDNA records could be used to depict temporal changes in population size of the early-spring Pacific-generated population and the Pacific stock, but not of the Inland Sea-generated subpopulations that mainly reside in the central Seto Inland Sea. This finding was further supported by a negative phase relationship between anchovy sedDNA and landings from the central Seto Inland Sea before 2000 (Supplementary Fig. 7).

Sardine and jack mackerel showed high landings in the Bungo Channel, with sedDNA for each species showing consistent temporal patterns with landings in the Bungo Channel, as well as in Japan (Supplementary Figs. 8 and 9). Therefore, temporal changes in sedDNA concentrations for these species reflect changes in abundance in the Bungo Channel. Since the channel and adjacent offshore waters represent one of the spawning areas for the Pacific stock of these species, population size is likely linked to the size of the Pacific stock for these species.

SedDNA from size fractions of 63–180 and 180–500 µm that included pellets (Supplementary Figs. 13–16) was not detected, or was detected in one of four replicates, with means $0.9 \pm 1.7$ and $0.3 \pm 0.6$ copies $g^{-1}$ dry sediment before sieved for sardine and jack mackerel, respectively (Table 2). However, during the time periods with abundant bulk DNA of these species, several samples had 20–120 copies $g^{-1}$ for sardine and 20–170 copies $g^{-1}$ for jack mackerel. These large values might not be explained by pellet-derived DNA alone. Moreover, anchovy DNA was not detected in all samples, despite having more abundant DNA in bulk sediments and larger stock abundance in the Seto Inland Sea than sardine and jack mackerel. Therefore, coarse sediment particles with >63 µm size fraction might not be the main source of fish DNA in bulk sediment. A previous study detected large quantities of DNA in particle size fractions ranging from 1 to 10 µm in aquatic samples[42]. These small particles might sink to the bottom via various sedimentation processes, and contribute a major source of fish sedDNA.

DNA was not detected in pore water samples in our study (Table 1). Thus, vertical DNA movement through pore water might be negligible. Leaching from sediments has mainly been observed in coarse-textured or unsaturated deposits that allow fluid advection across strata[43,44], and is generally regarded as a minor issue in permanently water-saturated sediments[3]. However, upward fluid advection in the surface layer could occur constantly, because of successive compaction associated with sedimentation across years, which might enhance DNA leaching. Our results indicate the absence of DNA in pore water, even in the uppermost 5 cm layer. We, therefore, concluded that DNA leaching in the sediment of Beppu Bay was negligible.

DNA was not detected from bones in the sediments. In contrast, fish scales yielded DNA, but only for anchovy ($1.5 \pm 4.2$ copies per scale). Given the maximum number of anchovy scales in dry bulk sediment (1.08–1.59 $g^{-1}$ based on data from[17]), a maximum DNA concentration derived from anchovy scales was estimated as ~11 copies $g^{-1}$ in dry bulk sediment. This value is two-orders of concentration lower than the 7-core mean concentration in DNA from dry bulk sediment (Supplementary Table 1).

What is biological quantity represented by sedDNA concentrations is complicated at present. The eDNA concentrations in water are positively correlated with the number of individuals and biomass of a species[24,25], thus eDNA might represent them[24,26], while eDNA release rate of fish is higher for the adult ones than for the juvenile ones[45] and higher release rates are observed during the spawning period[46,47]. Therefore, eDNA concentrations in water may not simply represent the number of individuals or biomass but also size composition of a population (i.e., juvenile/adult ratio) and the spawning behavior. However, contribution of spawning-related DNA release to the sedDNA may be minor for sardines, because sardine egg densities are low or zero in the Seto Inland Sea including Beppu Bay[48]. In addition, sardine larvae densities are also low or zero in the sea[48]. Therefore, sardine sedDNA may mainly represent the number of individuals or biomass for juvenile and adults. For anchovy, egg productions and larvae densities are high in the Seto Inland Sea and the spawning period is long (May to September)[48]. Therefore, egg productions and larvae biomass might be one of the factors controlling anchovy sedDNA concentration.

In conclusion, we detected sedDNA of Japanese anchovy, Japanese sardine, and jack mackerel in 1-m sediment sequences spanning from the present to 300 years ago, demonstrating the potential of obtaining the DNA of marine fish from marine sediments. Observed decadal-scale changes in the DNA concentrations of anchovy, sardine, and jack mackerel were consistent with those of landings in Japan and the Bungo Channel. Sardine DNA concentrations showed a significant positive relationship with fish scale abundance. Therefore, sedDNA concentrations may record decadal- and centennial-scale changes in fish species abundance in the water. DNA copies in the bulk sediment had much higher values than those of fish scales, bones, and pellets in the sediments; therefore, the DNA of fish species in the sediments is an independent proxy of fish abundance. The absence of vertical migration of pore water DNA indicates that the temporal succession of our DNA record was not disturbed. Thus, sedDNA represents a viable technique to track past changes in fish abundance, and could be used to reconstruct the abundance of macro-organisms inhabiting water.

## Methods

**Sediment cores for reconstructing variation in sedDNA**. To reconstruct temporal variation in fish DNA concentrations in bulk sediment (Experiment (a) in

Fig. 1), sediment core samples were collected from the deepest area of Beppu Bay (Supplementary Fig. 1b) on a survey vessel "Isana", Ehime University. All core tubes were cleaned with 0.6% sodium hypochlorite, tap water, and Milli-Q water. We collected three 120-cm-long gravity core samples (BG17-1, BG18-6, and BG18-8) (Supplementary Fig. 12) by using a 120-cm-long gravity corer (model HRL, RIGO Co. Ltd, Saitama, Japan; core ID was denoted as BG). Since gravity core sampling consistently fails to retrieve the surface layer, we collected three undisturbed core samples with a 60-cm-long multiple corer (Ashra, RIGO Co. Ltd, Saitama, Japan) (BMC17 S1-7, BMC17 S1-10, and BMC18-6 in Supplementary Fig. 11).

All of the cores consisted primarily of hemipelagic silty clayey sediments and a few millimeters or centimeters thickness-event layers with high-density, high-magnetic susceptibility, and coarser grains than those in hemipelagic layers (i.e., turbidites, which was probably formed by flooding and earthquake) (Supplementary Figs. 11 and 12). Since the sediment stratigraphy and chronology at the deepest site in Beppu Bay is well-known[29], these event layers could be correlated between cores at the deepest site in Beppu Bay by lithological features and stratigraphy on visual inspection, CT images, and magnetic susceptibility[29]. The age of these event layers in the sediment (Supplementary Table 4) was determined by previous studies[29], and were used as time markers and for dating the core samples used in this study (for details, see "Age determination" in Supplementary Methods).

Sediment cores were transferred to a pipe vertically split in half and pared in advance, and vertically split in half for subsequent processes to examine lithological and stratigraphic features. A few millimeter layer disturbed during the splitting process was carefully removed from the entire surface of the split core samples using a knife. Split cores were sectioned at intervals of every 1 or 2 cm for both working and archive core splits using cutting apparatus, and were transferred to lightproof bags and frozen at −80 °C. Pipes, knifes, and cutting apparatus were cleaned with 0.6% sodium hypochlorite, tap water, and Milli-Q water. To test an effect of time intervals during chilled preservation until subsampling on DNA preservation only the core BMC17 S1-7 was sectioned vertically at an interval of every 1 cm, after removing the margin of a sample cylinder. The section was transferred to lightproof bags, frozen soon after core collection during the vessel survey, and put in the freezer at −80 °C soon after the survey. Other core samples collected in 2017 were chilled for 4 weeks before being split, sectioned, and frozen at −80 °C. Core samples collected in 2018 were chilled for four days before being split and sectioned, and frozen at −80 °C. All core samples were kept at room temperature for <48 h before being frozen for CT scanning, measuring magnetic susceptibility, and sectioning processes. We used 1-cm- or 2-cm-thick samples at an interval of every 4 cm for the DNA analysis (Experiment (a) in Fig. 1). We analyzed 11, 12, 27, 26, 28, and 28 samples for BMC17 S1-7, BMC17 S1-10W (W: working split), BMC18-6W, BG17-1W, BG18-6W, BG18-8A (A: archive split), and BG18-8W, respectively, totaling 132 samples.

To examine contamination due to splitting, sectioning, subsampling, DNA extracting, and qPCR processes (Experiment (a) in Fig. 1) for marine fish sedDNA, lake sediment core samples were collected from Lake Biwa, Japan, with a 60-cm-long HR-type gravity core sampler (model HR, RIGO Co. Ltd, Saitama, Japan; core ID was denoted as BHR) (Experiment (a) in Fig. 1). We performed the same procedures as with the Beppu Bay samples and did in the same laboratories.

**Samples used to explore the source materials of sedDNA.** If there is a significant contribution of DNA from fossil fish scales, pellets, or bones, sedDNA signatures would reflect the concentrations of fish scales, pellets, or bones. Furthermore, if DNA in pore water contributes significantly to bulk sediment DNA, the vertical movement of DNA through pore waters would disturb historical abundance signatures of species. To examine whether sedDNA could be used as an independent index for detecting the population dynamics of different species, DNA concentrations of sedimentary fish scales, pellet size particle fractions, bones, and pore water were quantified by qPCR (Experiment (b) in Fig. 1). Sediment core samples collected from the deepest site in Beppu Bay (core BHR17-1 collected by HR-type undisturbed gravity core sampler) were divided into three fractions, fine and coarse particle fractions (63–180, 180–500 μm) and fish scales and bones (>500 μm) (Experiment (b) in Fig. 1). Size fraction samples were obtained using sieves with 63, 180, and 500 μm openings by washing them with tap water. We observed the material in each fraction (mainly composed of diatom valves, pellets, plant material, and clastic materials) under a microscope (Supplementary Figs. 13–16). Other sediment samples (BMC18-2 and BMC18-4) collected by using the Ashra-type multiple corer were sectioned at 5 cm intervals. The wet samples were divided into pore water and residual bulk sediments, and were centrifuged at 10,000 × g for 10 min at room temperature (Experiment (c) in Fig. 1).

**Extraction and purification of sedDNA.** DNA was extracted from sediment samples following the methods of a previous study[49]. In brief, 3 g sediment samples were incubated at 94 °C for 50 min in 9 ml alkaline solution composed of 6 ml of 0.33 M sodium hydroxide and 3 ml Tris-EDTA buffer (pH 6.7). After being centrifuged at 10,000 × g for 60 min, 7.5 ml of supernatant of the alkalized mixture was neutralized with 7.5 ml of 1 M Tris-HCl (pH 6.7). After adding 1.5 ml of 3 M sodium acetate (pH 5.2) and 30 ml absolute ethanol, the solution was preserved at −20 °C for more than 1 h. It was then centrifuged at 10,000 × g for 60 min. The pellet was transferred to a Power Bead tube that was installed in a fecal-soil DNA extraction kit (PowerSoil DNA Isolation Kit, MO Bio Laboratories, USA).

The "Experienced User Protocol 3 to 22" of the PowerSoil DNA Isolation Kit was followed. Finally, 100 or 200 μl DNA solution was obtained, and stored at −20 °C until quantitative PCR. Quantitative PCR and spike test methods that produced ΔCt (evidence of PCR inhibition) are described in the Supplementary Methods.

**Biogeochemical and geochemical analysis.** To explore the factors driving temporal variation in sedDNA concentration, we performed biogeochemical and geochemical analysis to obtain the concentration of biogenic opal, TOC, total nitrogen (TN), and Ti (Experiment (d) in Fig. 1; for details, see Supplementary Methods). Using the same samples used to analyze fish DNA concentrations (cores BMC17 S1-10 and BG17-1), biogenic opal analysis was conducted by removing carbonates and organic material, extracting biogenic opal, and molybdate-yellow spectrophotometry. TOC and TN concentrations of the BMC17 S1-10 and BG17-1 cores were determined using an HCN elemental analyzer for acid-washed and dried samples. We reported TOC and the molar ratios of TN:TOC (C/N ratios). A coefficient of variation for biogenic opal concentration of replicates of three samples was 2%. Replicate measurements of internal standards run with TOC and TN yielded coefficients of variation of 4.4% and 6.9%, respectively. Ti content in the archive halves of the BMC17 S1-11 (almost the same stratigraphy as all levels of BMC17 S1-10, because these cores were simultaneously collected by the multiple corer) and BG17-1 cores were measured using a micro-X-ray fluorescence spectrometry core scanner (ITRAX; COX Analytical Systems)[50,51] at the Center for Advanced Marine Core Research of Kochi University. We reported water content corrected Ti content calculated by ln-ratio of the Ti counts s$^{-1}$ with a Geometric mean of counts s$^{-1}$ of all high precision elements[52–54].

**Statistics and reproducibility.** For GAMs and Type II regression models, we converted uneven time series of sedDNA and fish scales to annual data by linear interpolation for all cores. Then, we calculated running means and errors for contemporary 7-year annual data of the multiple cores. We used a 7-year running mean for each core data for sedDNA and fish scales, because the fish scale data were previously analyzed at a time resolution of around 7 years by collecting samples at every 2-cm intervals in the sediment cores. All statistical analyses were performed in R ver. 3.5.1 (ref. [55]). To detect non-linear dynamics in DNA concentration data, we performed GAMs for the DNA concentration dynamics in all fish species with Gaussian distribution using the "gam" function in "mgcv" ver. 1.8.28 package. We performed Type II regression models with the standardized major axis method, with Gaussian distributions for the relationship between the DNA concentration of fish species with the fishery landing and fish scale concentrations. Gaussian Type II regression models were used, because evaluation of the preliminary model obtained higher $R^2$ values for Type II regression models with Gaussian distribution than for models with logarithmic distributions in all cases when using the "sma" function of "smatr" package ver. 3.4-3. We also performed Type II regression models for the relationship between the log$_{10}$-transformed values of DNA concentration of fish species and the fishery landing or fish scale concentrations.

To explore potential environmental factors driving temporal variation in sedDNA concentrations, we performed Pearson's correlation analysis between anchovy DNA copies with TOC content, molar carbon and nitrogen ratios (C/N), biogenic opal content, titanium content, and sedimentation rates (Supplementary Table 3). Lithological facies alternate from the massive lower layers to laminated upper layers (Supplementary Figs. 11 and 12), indicating changes in sedimentary and/or marine environments after 1960. Thus, we conducted a Pearson's correlation analysis between anchovy DNA and the stated parameters for the respective upper and lower layers. Analyses and t-tests were performed in Excel, and a significant level of 0.05 was used.

**Reporting summary.** Further information on research design is available in the Nature Research Reporting Summary linked to this article.

## Data availability
Source data are provided with this paper. All other data, if any, are available upon reasonable request.

## Code availability
The R codes used for data analyses during the current study are also available in Supplementary Data 1.

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

## Acknowledgements

We thank Hidejiro Onishi, Ehime University, for conducting sampling with E/R/V Isana and Yukiko Goda, the Center for Ecological Research, Kyoto University for conducting sampling with R/V Hasu. We also thank Mariko Nagano, University of Hyogo, for supporting the laboratory experiments. We thank Hiroki Nakao for providing local fish landing data. This study was supported by Grants-in-Aid for Scientific Research (17K20045, 19H04284) from the JSPS, the Environment Research and Technology Development Fund (4-1602) of Environmental Restoration and Conservation Agency, Japan and JST-CREST (JPMJCR13A2), and Asahi Group Foundation in 2017. The cooperative research program (17A065, 17A046, and 18A024) of the Center for Advanced Marine Core Research, Kochi University also supported this study.

## Author contributions

M.K. and H.D. conceived and equally contributed to the study. M.K., H.T., and Y.S. conducted sampling and experiments; H.D. conducted some of the experiments, performed data analyses of DNA/fish scale/landings, and wrote this section of the manuscript; T.M. and M.K.S. developed the methodology of sedDNA extraction and wrote this part of the manuscript; M.K. and H.T. wrote the first draft of the manuscript; H.D. and T.M. revised the manuscript. All authors commented on the revised manuscript.

## Competing interests

The authors declare no competing interests.
