## [Peer Review File · Communications Biology]

Reviewers' comments (first review):

Reviewer #1 (Remarks to the Author):

sedDNA has been a very popular tool in recent years to discover the footprints of the past biodiversity, though sedDNA can be affected by various environmental factors and the dynamic has not been fully understood, particularly for macro-organisms. The authors attempted to use sedDNA tracking the decadal-centennial dynamics of three dominant marine fish species. They finally revealed the changes in three marine fishes' abundance over 300 years. This certainly is a very interesting study, while the improvement or innovation of this study is not obvious compared with previous studies. Only applying the sedDNA approach into so-called macro-organisms (like fishes) might not be enough. In addition, some parts should be explained with detailed information, and the authors should consider the following points.

1. As most of results are also in similar studies (e.g., plants, invasive species), what's the major contribution to eDNA approach or protection and utilization of fishery resources? Extensions on methods/applications alone might not be enough to show the innovation. The value of eDNA or sedDNA in fishery management and fishery resource utilization should be highlighted.
2. Many of the references cited in this study are freshwater ecosystems, however, the preservation, release and degradation of DNA in sediments can be completely different between marine and freshwater environments (e.g., pH and light). Do these freshwater findings apply to the marine?
3. Figure 1, changes in DNA concentration of three fish species are very high in the surface layer and very low in the bottom layer. Was this related to DNA degradation? How to quantitatively separate the factor of DNA degradation cross the past 300 years? It might be hard to obtain quantitative information about fish decades or centuries ago. Thus, to characterize the presence or absence of fish using sedDNA is more reliable than quantifying it.
4. The part of "Age determination" is hard to follow, is this from the refers or your measurement? Please rephrase.
5. Some part (including Target fish species and Quantitative PCR) can be put into SI.
6. There are some expressions which would need further explanation as they make the paper hard to follow. Specifically see:

-Line 23. “abundance”. Which or What?

-Line 30. “fish DNA concentrations”. Which one?

-Line 60. “FAO”. Provide the full name?

-Line 69-70. “DNA concentration”. Total DNA or fish DNA?

-Line 122. “negative control”. Including DNA extraction, PCR processes? Cannot find the description in the full text.

-Line 353. “an interval of every 1 cm or 2 cm”. Why not keep unity?

Reviewer #2 (Remarks to the Author):

Kuwae and colleagues evaluated how qPCR quantification of sediment-preserved DNA might reflect historic biomass changes of fish species, in comparison with scale counts, and historic records of catch.

At the very beginning I want to emphasize that this and similar studies are extremely important to study historic biodiversity dynamics: I am personally convinced that ancient environmental DNA is one of the very few data sources that can provide really long-term, relatively good resolution data on species presence and possibly population sizes. The authors are perfectly right that of these two, it is rather the historic population size estimates that receives less consideration, and this is exactly what they are targeting.

I strongly recommend that the authors use this amazing dataset and put way more effort to properly analyze and interpret it. My most serious problem is that the conclusions reached and interpretations simply do not reflect the results. The authors claim in the abstract that “Temporal changes to fish DNA concentrations were consistent with those of fish scale abundance and landings from neighboring areas and in Japan. Thus, sedimentary DNA could be used to track decadal-centennial dynamics of fish abundance in marine waters”. In contrast, most of the results show that DNA data sometimes corresponds with catch data and/or scale data, and at other times not at all. I agree that the correspondence suggests that eDNA can be used as historic biomass indicators in SOME cases. However, the authors unfortunately miss the opportunity to properly evaluate the cases when the

correlations do not work out - and understanding the limitations of eDNA is really important when it is

overall advertised as the ultimate solution for many data-hungry biodiversity fields. In this respect I was surprised to see no sediment chemistry in the manuscript: enzymatic inhibition is probably the best candidate for mismatches between DNA and other detection methods, and this might be inferred from sediment properties. I don't remember even seeing enzymatic inhibition mentioned as a possible explanation for failing correlations. qPCR curves should contain some indication of inhibition. Again - it is extremely important for the development field to understand when eDNA methods work and when not. I urge the authors to contribute to this understanding by reconsidering their data, to clearly define where they bring new understanding, and to give forward-leading recommendations for what needs to be done next to understand historic sediment eDNA.

Methods:

- The description of the methods is not easy to get at places.
- I could not locate the data and the analysis script, although it is checked in the reviewer form that it is available for review. It is important that the authors make both the data and the analysis script available for review, and that they deposit them after publication in one of the data/script sharing platforms.
- This MS will much benefit a conceptual figure of the work process. This figure should explain the coring setup, the sub-sampling of the cores, DNA extraction for different purposes (historic biomass evaluation, fishscale and porewater etc DNA, etc.), sediment analyses, qPCR setup. Currently I think I don't understand what was exactly done. Explaining the process of such a complex experiment exclusively in text is generally very difficult.

Discussion:

- In general, the possible reasons behind the results are mostly unexplored, or poorly argued.
- If I get it right, all sediment cores come from the same place. This is an excellent opportunity to say something about the representativeness of eDNA results in any single core. I think it is one of the major unknowns with historic eDNA how well a few grams of sediment from a horizon of a single core represent huge communities (or populations).

206-212: Hard to follow. May the primers preferentially amplify Anberstripes?

215-216: I find this an extremely important result for the field. This should be developed and interpreted more: vertical transport of DNA in sediments is debated a lot, and currently the best argument for the lack of this transport is that sediments are water-saturated, so there is no reason for DNA movement (which I find as a plausible, but "soft" argument). The fact that there is actually no

DNA in porewater is actually a hard argument for the lack of vertical transport.

217-224: I am lost: what samples were used for DNA extractions if sardine and mack. were found only in 3 bulk samples altogether? Draw a conceptual figure of the entire work process.

347-349: I count seven cores on many figures, here are mentioned six.

364-368: Explain your negative control procedure in much more detail.

378, 384-385: how did you control for contamination when separating sediments into five fractions, to explore fish DNA sources?

391: Probably start with explaining that the sediment stratigraphy and chronology at the site is well-known (with references to the papers that publish this), and then say that you just correlate the cores with event layers.

417-426: What extracellular DNA fraction does the extracted sedDNA correspond with? See Torti, A., Lever, M. A., & Jørgensen, B. B. (2015). Origin, dynamics, and implications of extracellular DNA pools in marine sediments. *Marine Genomics*, 24, 185–196. <https://doi.org/10.1016/j.margen.2015.08.007>

Figures in general: Tell the message of the figure: what should we understand from them? - before the technical description of the objects.

Fig. 1. I would use the recommended experimental design figure as the 1st figure, and put this Fig. 1 into the supplement. How do the different cores match if you 7-year average them?

Fig. 2. For easier understanding I suggest to do a grid plot, i.e. `method ~ species` in R `ggplot` with `facet_grid`

Fig. 4d. I don't get it why mackerel DNA should correspond with mackerel + amberstripe, but not with mackerel.

Rebuttal letter (first review)

Thank you for your response for our paper. We gratefully thank anonymous reviewers for providing thoughtful comments and ideas that greatly improved the manuscript. We revised the manuscript totally according to their comments. Please find our responses (described after [Response]) to their specific comments.

The main revisions we made are (1) we addressed the major contribution of the sedDNA technique to protection and utilization of fishery resources (response to Reviewer #1), (2) according to comments from Reviewer #2 who suggests that we do not mentioned sediment chemistry in the manuscript and enzymatic inhibition, we added other sediment chemistry data and results of “spike test” for detection of effects of enzymatic inhibitors to support our conclusion, the utility of sedDNA concentrations to elucidate species population dynamics, (3) we added figure of experimental design to make clear our entire work processes (Fig. 1) (response to Reviewer #2), and (4) we deposited our data and analysis script used in the figures and tables in the tentative sharing data repository as the following address: <https://> (response to Reviewer #2). We will deposit them in the ‘figshare’ after publication.

Accompanied by adding the sediment chemistry data, we added a coauthor, Yoshiaki Suzuki, who has analyzed it.

Best regards,
Michinobu Kuwae

Reviewers' comments (first review):

Reviewer #1 (Remarks to the Author):

sedDNA has been a very popular tool in recent years to discover the footprints of the past biodiversity, though sedDNA can be affected by various environmental factors and the dynamic has not been fully understood, particularly for macro-organisms. The authors attempted to use sedDNA tracking the decadal-centennial dynamics of three dominant marine fish species. They finally revealed the changes in three marine fishes' abundance over 300 years. This certainly is a very interesting study, while the improvement or innovation of this study is not obvious compared with previous studies. Only applying the sedDNA approach into so-called macro-organisms (like fishes) might not be enough. In addition, some parts should be explained with detailed information, and the authors should consider the

following points.

1. As most of results are also in similar studies (e.g., plants, invasive species), what's the major contribution to eDNA approach or protection and utilization of fishery resources? Extensions on methods/applications alone might not be enough to show the innovation. The value of eDNA or sedDNA in fishery management and fishery resource utilization should be highlighted.

[Response] We addressed the value of sedDNA in fishery management and resource utilization in the introduction (L.50-66).

2. Many of the references cited in this study are freshwater ecosystems, however, the preservation, release and degradation of DNA in sediments can be completely different between marine and freshwater environments (e.g., pH and light). Do these freshwater findings apply to the marine?

[Response] The reviewer's comments are very good point for generalization of this sedimentary DNA technique applied for the relevant researches. Our assertion about potential utility of sedimentary DNA described in the introduction indeed relies on findings from freshwater studies in relation to preservation and degradation in sediments. And, we have no reference about sedimentary DNA preservation potential for marine macro-organism, so we could not refer suitable references. However, our results clearly show evidences of the long-term DNA preservation in marine sediments as seen in freshwater sediments. Therefore, preservation potential of DNA in sediments might have common mechanisms in both marine and freshwater systems. In this point, several previous studies from marine and freshwater systems indicate that adsorption of extracellular DNA on clay minerals and organic compounds such as humic substances and proteins likely play an important role in stability and preservation potential of DNA; clay minerals and organic compounds exist abundantly in both marine and freshwater bottom environments. Bottom environments are completely different between marine and freshwater systems as suggested by the Reviewer, but, our finding supports preservation potential of DNA in sediments in marine systems as well as freshwater systems and utility of sedimentary DNA to track population dynamics as well as absence/presence of species in waters. Therefore, we do not make any changes in the introduction, but add a part of this description in discussion in the revised version (L.279-285).

3. Figure 1, changes in DNA concentration of three fish species are very high in the surface layer and very low in the bottom layer. Was this related to DNA degradation? How to quantitatively separate the factor of DNA degradation cross the past 300 years? It might be hard to obtain quantitative information

about fish decades or centuries ago. Thus, to characterize the presence or absence of fish using sedDNA is more reliable than quantifying it.

[Response] Abnormal high concentrations in organic materials in the surface layer derived from fresh labile organic matters (e.g. chlorophyll a etc.) can be seen in many organic biogeochemical studies; the very high DNA concentrations (BMC18-6, BMC17 S1-7) may also be the case. Enriched DNA concentrations in the surface layer is probably formed by the recent deposition during the month or more before we collected, the surficial organic materials may be susceptible to rapid decomposition due to early diagenesis in a few years. To assess the effect of early diagenesis on surface DNA enrichment may require further study. However, except for the surface layer we can find decadal to centennial variability in fish DNA abundance. In especial, DNA abundance for sardine were higher in the 18th century than in the latter half of 20th, suggesting that diagenesis of fish DNA in the Beppu Bay sediments does not significantly deplete DNA concentrations on centennial timescales.

4. The part of “Age determination” is hard to follow, is this from the refers or your measurement? Please rephrase.

[Response] We rephrased this part (L.445-451, and L.15-29 in ‘Age determination’ in Supplementary information).

5. Some part (including Target fish species and Quantitative PCR) can be put into SI.

[Response] We put these parts into SI (L.5-13 in ‘Target fish species’ and L.31-98 in ‘Quantitative PCR, both in the Supplementary Information.

6. There are some expressions which would need further explanation as they make the paper hard to follow. Specifically see:

-Line 23. “abundance”. Which or What?

[Response] We added “ of a species” (L.26).

-Line 30. “fish DNA concentrations”. Which one?

[Response] We modified this sentence (L.32-34).

-Line 60. “FAO”. Provide the full name?

[Response] We added the full name (L.70).

-Line 69-70. “DNA concentration”. Total DNA or fish DNA?

[Response] We added “fish” (L.78-80).

-Line 122. “negative control”. Including DNA extraction, PCR processes? Cannot find the description in the full text.

[Response] We added “for sectioning, subsampling, DNA extraction, and PCR processes” (L.137-138)

-Line 353. “an interval of every 1 cm or 2 cm”. Why not keep unity?

[Response] We analyzed 2cm-thick samples only for BG18-6W because we need to assess how the number of a laminae which was formed in a season influence sedDNA concentrations (more sets of laminae cause lower concentrations than less sets of laminae if there are seasonal laminae with zero values of DNA concentrations which is expected in winter-formed laminae due to absence of anchovy in Beppu Bay). However, we did not find significant depletions of DNA concentrations for 2cm-thickness samples of BG18-6W (Supplementary Figure 3).

Reviewer #2 (Remarks to the Author):

Kuwae and colleagues evaluated how qPCR quantification of sediment-preserved DNA might reflect historic biomass changes of fish species, in comparison with scale counts, and historic records of catch.

At the very beginning I want to emphasize that this and similar studies are extremely important to study historic biodiversity dynamics: I am personally convinced that ancient environmental DNA is one of the very few data sources that can provide really long-term, relatively good resolution data on species presence and possibly population sizes. The authors are perfectly right that of these two, it is rather the historic population size estimates that receives less consideration, and this is exactly what they are targeting.

I strongly recommend that the authors use this amazing dataset and put way more effort to properly analyze and interpret it. My most serious problem is that the conclusions reached and interpretations simply do not reflect the results. The authors claim in the abstract that “Temporal changes to fish DNA concentrations were consistent with those of fish scale abundance and landings from neighboring areas and in Japan. Thus, sedimentary DNA could be used to track decadal-centennial dynamics of fish abundance in marine waters”. In contrast, most of the results show that DNA data sometimes corresponds with catch data and/or scale data, and at other times not at all. I agree that the correspondence suggests that eDNA can be used as historic biomass indicators in SOME cases. However, the authors unfortunately miss the opportunity to properly evaluate the cases when the correlations do not work out - and understanding the limitations of eDNA is really important when it is overall advertised as the ultimate solution for many data-hungry biodiversity fields. In this respect I was surprised to see no sediment chemistry in the manuscript: enzymatic inhibition is probably the best candidate for mismatches between DNA and other detection methods, and this might be inferred from sediment properties. I don't remember even seeing enzymatic inhibition mentioned as a possible explanation for failing correlations. qPCR curves should contain some indication of inhibition. Again - it is extremely important for the development field to understand when eDNA methods work and when not. I urge the authors to contribute to this understanding by reconsidering their data, to clearly define where they bring new understanding, and to give forward-leading recommendations for what needs to be done next to understand historic sediment eDNA.

[Response] We agree with Reviewer's comments: we indeed miss the opportunity to properly evaluate the cases when the correlations do not work out. The notable discrepancy is seen between anchovy DNA and fish scale concentrations. According to his comments, we performed spike test for effects of enzymatic inhibition on PCR amplifications. However, the results showed

no enzymatic inhibition for all the extracted DNA solutions from the sediment samples used for historic variations in sedDNA (all ΔC_t values were less than 3 cycles) (we added descriptions about this in the revised manuscript (L.144-146, L.291-296 and L.513-514) and in Supplementary information (L.94-98)). We also added the sediment chemistry data (Experiment (d) in Fig. 1) and confirmed that there is no effect of inputs of terrestrial organic matters including abundant humin substances (one of the main enzymatic inhibitors) on DNA sedimentation as suggested by no significant correlation between anchovy DNA and C/N ratio (see L.296-298 and for detail, see ‘Environmental factors driving sedimentary DNA concentrations’ in the Supplementary information, L.127-196). In addition, we confirmed that there is no significant effect of inputs of other enzymatic inhibitor, soil minerals (clay), on DNA sedimentation as suggested by no significant correlation between anchovy DNA with sedimentation rates and a weak negative correlation between anchovy DNA and Ti content (an index of soil mineral abundance). We described this in the text (L.296-298) and ‘Environmental factors driving sedimentary DNA concentrations’ in the Supplementary information (L.127-196). Accompanied by adding the Ti content data, we added a coauthor (Dr. Yoshiaki Suzuki) who has analyzed it.

We suspect that the discrepancy between DNA and fish scales results from more heterogeneous deposition rates of fish scales, which cause false peaks because of larger size than DNA-holding particles. In fact, anchovy fish scale record (Fig. 2c) failed to tracking a 1950-1970 peak seen in catch record (Fig. 2b), while DNA record track this peak (Fig. 2a). In order to clarify this issue, examinations on more sediment core DNA and scale data are required; we think this is what needs to be done next. Nevertheless, since there is a statistically significant relation between DNA concentrations and landings for all three species and between DNA concentrations with scale concentrations for sardine, our conclusion “sedimentary DNA records signatures of abundance of fish species and could be used to track decadal-centennial dynamics of fish abundance” may be reasonable. This is where we bring new understanding.

Methods:

- The description of the methods is not easy to get at places.
- I could not locate the data and the analysis script, although it is checked in the reviewer form that it is available for review. It is important that the authors make both the data and the analysis script available for review, and that they deposit them after publication in one of the data/script sharing platforms.

[Response] We uploaded our data and analysis script in the tentative sharing data repository <https://www.dropbox.com/sh/g9vd4pdjukzmp1k/AAD69VnaAbKGb3-xW5qQF9TEa?dl=0>

. We will deposit them in the figshare after publication.

- This MS will much benefit a conceptual figure of the work process. This figure should explain the coring setup, the sub-sampling of the cores, DNA extraction for different purposes (historic biomass evaluation, fishscale and porewater etc DNA, etc.), sediment analyses, qPCR setup. Currently I think I don't understand what was exactly done. Explaining the process of such a complex experiment exclusively in text is generally very difficult.

[Response] According to his comment, we added a conceptual figure of the entire work process (Fig. 1 in the revised manuscript) and addressed it in the introduction (L.100-113).

Discussion:

- In general, the possible reasons behind the results are mostly unexplored, or poorly argued.
- If I get it right, all sediment cores come from the same place. This is an excellent opportunity to say something about the representativeness of eDNA results in any single core. I think it is one of the major unknowns with historic eDNA how well a few grams of sediment from a horizon of a single core represent huge communities (or populations).

[Response] We appreciate this Reviewer's comment. We argued this point in detail in discussion (L.300-315). Beppu Bay is located near waters off south Japan which is main spawning areas of

the Pacific stock of Japanese sardine, which is the largest Japanese sardine stock. Growing areas of Japanese sardine shrank and extended on decadal timescales but main spawning areas have been stable within waters off south Japan.

Therefore, waters close to spawning area like Beppu Bay are most appropriate to produce a record of regional sardine abundances in the western North Pacific. In fact, year to year recruitments of sardine in the Seto Inland Sea including Beppu Bay show good correlation with those in main spawning areas (Zenitani et al., 2001, Bull. Jap. Soc. Fish. Oceanogr.). Therefore, sedDNA in a few gram sediment sample is expected to be able to track DNA sedimentation rates being proportional to sardine abundance in Beppu Bay, probably linking to the Pacific stock abundance. This is argued in Kuwae et al. (2017) in Progress in Oceanography.

Spawning areas of Japanese anchovy are located in the Seto Inland Sea and off south and

Fig. 10. Migration areas of Japanese anchovy (*Engraulis japonicus*) stocks (Pacific stock, Seto Inland Sea stock, and Tsushima Warm Current stock) (lower window) and spawning grounds of the inland-sea-generated population (A: bold gray line area, i.e., the Seto Inland Sea) and the Pacific-generated population (B: bold dashed line area) and generating seasons and migration routes of subpopulations of the Seto Inland Sea stock (upper panel). The migration areas of the respective Japanese anchovy stocks and spawning grounds and routes of the Pacific-generated population and inland-sea-generated population are based on descriptions in Takao (1964), Fisheries Agency and Fisheries Research Agency of Japan (2005), and Asami (1962).

northeast Japan; there are the Seto Inland Sea stock and the Pacific stock. Catch data shows different temporal patterns between that the Seto Inland Sea stock and the Pacific stock. But, decadal-scale changes in catch for the Pacific stock are similar within south and northeast Japan except for after 1990. Therefore, decadal changes in the anchovy population size are seen in the broad areas off Japan. Our DNA record shows a similar decadal changes to catch in the Pacific or the Bungo Channel, thus anchovy sedDNA may track temporal changes in anchovy populations originated from the Pacific stock. We described this in the text (L.364-377).

206-212: Hard to follow. May the primers preferentially amplify Anberstripes?

[Response] We modified this sentence for readers to more easily follow (L.232-240).

215-216: I find this an extremely important result for the field. This should be developed and interpreted more: vertical transport of DNA in sediments is debated a lot, and currently the best argument for the lack of this transport is that sediments are water-saturated, so there is no reason for DNA movement (which I find as a plausible, but “soft” argument). The fact that there is actually no DNA in porewater is actually a hard argument for the lack of vertical transport.

[Response] We appreciate this Reviewer’s comment. We added more discussion on this issue (L.398-406) and move the previous supplementary table for the pore water experiments into

Table 1 in the main manuscript because of its significance of our results.

217-224: I am lost: what samples were used for DNA extractions if sardine and mack. were found only in 3 bulk samples altogether? Draw a conceptual figure of the entire work process.

[Response] According to his comment, we added a conceptual figure of the entire work process and modified the relevant description in the text (L.265-270).

347-349: I count seven cores on many figures, here are mentioned six.

[Response] There are six cores for reconstruction of historical variations in bulk sedDNA. We vertically split in half for core BG18-8 and named them as BG18-8W (W: working split) and BG18-8A (A: archive split). Naming W or A often are used conventionally for paleoceanographic field. We added the information in the conceptual figure (Fig. 1) and the text (L.452-454, L.469-472).

364-368: Explain your negative control procedure in much more detail.

[Response] To test whether there is contamination during processes of splitting, sectioning, subsampling, DNA extraction, and qPCR for the Beppu Bay samples, we conducted the same procedures for Lake Biwa sediment samples in which there must be no marine fish DNA (see L.473-478 and Fig. 1a).

378, 384-385: how did you control for contamination when separating sediments into five fractions, to explore fish DNA sources?

[Response] We used stainless steel sieves for separating fraction with tap waters. Sieves were cleaned with 0.6% sodium hypochlorite to remove contaminations. We didn't analyze tap waters during the DNA source analyses, but we believe that tap water is DNA-free. We don't think that there is significant DNA contamination from less than 64um size particles for sieve-separated size fraction samples and fish scales because we confirmed no the size particles and scales at visual inspections by a binocular. However, even if it happened, our results (fish DNA concentrations in the coarser size fractions and in fish scales were much less than those in the <64um size fraction) do not change.

391: Probably start with explaining that the sediment stratigraphy and chronology at the site is well-

known (with references to the papers that publish this), and then say that you just correlate the cores with event layers.

[Response] We appreciate his comment. We modified sentences according to the comments (L.445-451).

417-426: What extracellular DNA fraction does the extracted sedDNA correspond with? See Torti, A., Lever, M. A., & Jørgensen, B. B. (2015). Origin, dynamics, and implications of extracellular DNA pools in marine sediments. *Marine Genomics*, 24, 185–196. <https://doi.org/10.1016/j.margen.2015.08.007>

[Response] After incubated at 94 °C for 50 min in 9 mL alkaline solution and centrifuged, it is expected that the supernatant includes intracellular DNA (structurally intact dead cells) and extracellular DNA which was adsorbed on to sediment minerals and originated from insoluble detrital organic/inorganic components or locked inside sediment aggregates. The final solutions after DNA extraction using PowerSoil DNA Isolation Kit may include dissolved DNA derived from all those structures.

Figures in general: Tell the message of the figure: what should we understand from them? - before the technical description of the objects.

[Response] We changed the captions according to Reviewer's comment.

Fig. 1. I would use the recommended experimental design figure as the 1st figure, and put this Fig. 1 into the supplement. How do the different cores match if you 7-year average them?

[Response] We replaced Fig. 1 to experimental design, and put the previous Figure 1 into the supplementary information. We dated each core independently using time markers, including event 0a, 0b, and etc. (see in detail in the section 'Age determination' in the Supplementary Information) and converted uneven core data into annual data by linear interpolation for all the cores. Then we averaged for contemporary 7-yr annual data of the multiple cores (see in 'Data analysis' in the main text (L.538-541)).

Fig. 2. For easier understanding I suggest to do a grid plot, i.e. `method ~ species` in R `ggplot` with `facet_grid`

[Response] We replaced it into a grid plot for Fig. 2.

Fig. 4d. I don't get it why mackerel DNA should correspond with mackerel + amberstripe, but not with mackerel.

[Response] This is because duration of Jack mackerel time series is too short to figure out the relationship between the fish catch and the DNA concentrations. But the record of *Caranginae* landing (sum of jack mackerel and amberstripe scad) is enough to detect the relationship. Jack mackerel catches after 1960 accounted for $79 \pm 14\%$ of *Caranginae* landing, so we can use this data instead of mackerel (see L. 234-240).

Reviewers' comments (second review):

Reviewer #1 (Remarks to the Author):

This manuscript has been significantly improved from the previous version.

A revision would benefit from moving some figures from SI to the main text, for example S Fig 4, and S figure 5,

Reviewer #3 (Remarks to the Author):

This article uses qPCR to quantify the “abundance” (I would prefer the term “biomass”) of three fish species in marine sediments. This is an interesting paper that might constitute a primer in the quantification of past fish biomasses from sediments. I did not read the initial version of this MS, but I found that the comments of the two initial referees were well considered in this revised version. Although the relationships between sedDNA, scale density and landings are not really tight, I understand that the authors used the best available historical data. I strongly appreciate the effort to use two distinct proxies of population abundance (landings and scales), and to consider several species although the relationships with sedDNA and proxies of abundance are not always significant. Overall those relationships were well interpreted in the discussion and I finally found the paper convincing. I nevertheless remain unsure on the meaning of landings as a proxy of abundance. Therefore, the link between sedDNA and landings might only partly reflect the relevance of the sedDNA method (see details below). It is also a bit frustrating to see an excellent fit between scales and sedDNA for the sardine, but not for the anchovy, but the explanations given to explain this mismatch in the discussion provide some potential explanations.

Please find below my detailed comments.

1. This paper quantifies the abundance of fish DNA in sediments using qPCR. I am nevertheless unclear about the meaning of fish abundance here. If you mean the number of individuals, how can you sort adults, juveniles and larvae? I suppose eggs sedimentation to the sea bottom, and larval mortality contribute to the sedDNA, so what is a fish individual here? An easy way to solve this problem would be to consider that DNA abundance is a proxy of the fish biomass instead of fish abundance. I therefore suggest to replace “abundance” by “biomass” in the title and throughout the MS, or at least to provide a short discussion about the meaning of “abundance”.

2. The MS put a strong emphasis on the relationship between sedDNA abundance and fish landings but there is almost no information on fish landing data. How this data was collected? Is this data relevant?, Which is the fishing pressure and is this a relevant proxy of fish biomass/abundance?, Did the fishing pressure varied across time? I suspect that fishing methods evolved from the early 20th century to now, and how this affected landings? So did you used raw landing data or corrected values accounting for different sampling efforts?

3. Line 287-289: The argument that DNA degradation with time do not affect the conclusions because DNA abundance was higher in 1850 layer than in the 1980 layer implies that fish abundance remained stable across time, which is not verified. You indeed report changes in targeted fish sizes in recent decades (lines 340-350). Therefore you cannot exclude DNA degradation with time, and the abundance of older DNA deposits might underestimate the DNA abundance, this should at least be noticed in the MS.

4-Lines 325-329: The lack of relationship between scales abundance and landings of anchovy is here interpreted as proof that the scale abundance is not a good proxy of anchovy abundance. I do not understand this. Landings might also be a biased proxy of fish abundance (see above comment), so the assertion that "...sedDNA might track temporal variation in fish abundance in the overlying water to a greater extent than fish scale abundance, because the abundance of anchovy scales showed no significant correlation with catches in Japan" might not be true.

5- In the figures 3 and 4, regression lines and confidence intervals should not be represented for non-significant relationships.

6- The figures 3 and 4 show that significant relationships are driven by a few samples with high values. The variables should be log-transformed prior to the regression analysis to ensure a better contribution of the whole range of values to this relationship.

Rebuttal letter (second review)

Thank you for your response for our paper. We gratefully thank anonymous reviewers for providing thoughtful comments and ideas that greatly improved the manuscript. We revised the manuscript totally according to their comments. Please find our responses (described after [Response]) to their specific comments.

The main revisions we made are (1) we moved Supplementary Fig. 4 and S. Fig. 5 to the main text (response to Reviewer #1), (2) according to comments from Reviewer #3 who suggests that we do not make clear fish ‘abundance’, we defined ‘abundance’ as both ‘biomass’ and ‘the number of individuals’ in the introduction and discuss what is biological quantity represented by sedDNA in the last part of Discussion section, (3) we added more information on fish landing data we use (response to Reviewer #3), and (4) we performed the regression analysis using log-transformed sedDNA and landing data these results in the Supplementary Figs. 5 and 6.

Best regards,

Michinobu Kuwae

Reviewers' comments (second review):

Reviewer #1 (Remarks to the Author):

This manuscript has been significantly improved from the previous version.

A revision would benefit from moving some figures from SI to the main text, for example S Fig 4, and S figure 5,

[Response]

We moved S Fig. 4 and S Fig. 5 to the main text (Fig. 2 and Fig. 3, respectively).

Reviewer #3 (Remarks to the Author):

This article uses qPCR to quantify the “abundance” (I would prefer the term “biomass”) of three fish species in marine sediments. This is an interesting paper that might constitute a primer in the quantification of past fish biomasses from sediments. I did not read the initial version of this MS, but I found that the comments of the two initial referees were well considered in this revised version.

Although the relationships between sedDNA, scale density and landings are not really tight, I understand that the authors used the best available historical data. I strongly appreciate the effort to use two distinct proxies of population abundance (landings and scales), and to consider several species although the relationships with sedDNA and proxies of abundance are not always significant. Overall those relationships were well interpreted in the discussion and I finally found the paper convincing. I nevertheless remain unsure on the meaning of landings as a proxy of abundance. Therefore, the link between sedDNA and landings might only partly reflect the relevance of the sedDNA method (see details below). It is also a bit frustrating to see an excellent fit between scales and sedDNA for the sardine, but not for the anchovy, but the explanations given to explain this mismatch in the discussion provide some potential explanations.

[Response]

We appreciate his thoughtful comments. We basically revised our manuscript according to his comments.

Please find below my detailed comments.

1. This paper quantifies the abundance of fish DNA in sediments using qPCR. I am nevertheless unclear about the meaning of fish abundance here. If you mean the number of individuals, how can you sort adults, juveniles and larvae? I suppose eggs sedimentation to the sea bottom, and larval mortality contribute to the sedDNA, so what is a fish individual here? An easy way to solve this problem would be to consider that DNA abundance is a proxy of the fish biomass instead of fish abundance. I therefore suggest to replace “abundance” by “biomass” in the title and throughout the MS, or at least to provide a short discussion about the meaning of “abundance”.

[Response]

We appreciate this comment. The comment can help us to improve our MS. What is biological quantity represented by sedDNA may be a fundamental issue to be addressed in the next step. At present, we cannot use “biomass” instead of “abundance” because it is too difficult to logically describe “abundance” with discriminating biomass from the number of individuals in the entire MS. However, we define “abundance” as both the number of individuals and biomass in the introduction (L.48-49 in the Revised manuscript-marked up) and discuss its meaning of sedDNA in the last part of Discussion (L.415-431 in the Revised manuscript-marked up). At this point, we can only say that sedDNA is influenced by several components of population dynamics of a species, including not only biomass and the number of individuals especially through size

composition of a population (juvenile/adult ratio) but also egg productions.

2. The MS put a strong emphasis on the relationship between sedDNA abundance and fish landings but there is almost no information on fish landing data. How this data was collected? Is this data relevant?, Which is the fishing pressure and is this a relevant proxy of fish biomass/abundance?, Did the fishing pressure varied across time? I suspect that fishing methods evolved from the early 20th century to now, and how this affected landings? So did you used raw landing data or corrected values accounting for different sampling efforts?

[Response]

We added information on the fish landing data in the Methods section of the Supplementary Information (L.15-25), including collection of landing data (raw landing data without larvae), fishing pressure, and relevance to the proxy of fish biomass. We do not use corrected values accounting for fishing efforts. It is so much difficult to estimate catch per unit effort because collection methods of landing data differ between the intervals before and after 1950. Fisheries-oceanographic researchers point out that fishing efforts or pressure varied across time, including significantly decreased fishing efforts during the World War II, the underestimation of sardine landings during the 1930s-1940s sardine flourish interval due to fishing methods being less efficient than those during the 1980s sardine flourish interval, and increased fishing pressure to anchovy larvae after 1980. Such evolution of fishing affect landing data to bias stock biomass changes, as pointed out by Reviewer #3. Therefore, this bias most likely causes weak Type II relationships between sedDNA and landing. Nonetheless, our results show statistically significant relationships, which do not affect our conclusion.

3. Line 287-289: The argument that DNA degradation with time do not affect the conclusions because DNA abundance was higher in 1850 layer than in the 1980 layer implies that fish abundance remained stable across time, which is not verified. You indeed report changes in targeted fish sizes in recent decades (lines 340-350). Therefore you cannot exclude DNA degradation with time, and the abundance of older DNA deposits might underestimate the DNA abundance, this should at least be noticed in the MS.

[Response]

We appreciate his comments. sedDNA diagenetic degradation and the underestimation of older DNA deposition rates cannot completely be excluded. We added this information in the present manuscript (L.291-293 in the Revised manuscript-marked up).

4-Lines 325-329: The lack of relationship between scales abundance and landings of anchovy is here interpreted as proof that the scale abundance is not a good proxy of anchovy abundance. I do not understand this. Landings might also be a biased proxy of fish abundance (see above comment), so the assertion that "...sedDNA might track temporal variation in fish abundance in the overlying water to a greater extent than fish scale abundance, because the abundance of anchovy scales showed no significant correlation with catches in Japan" might not be true.

[Response]

We agree with his comments. Landings might also be biased due to temporal differences in fishing efforts. We removed this sentence (L.328-331 in the Revised manuscript-marked up).

5- In the figures 3 and 4, regression lines and confidence intervals should not be represented for non-significant relationships.

[Response]

We removed regression lines and confidence intervals from the panels which showed non-significant relationship (Figs. 5 and 6 and Supplementary Figs. 5 and 6).

6- The figures 3 and 4 show that significant relationships are driven by a few samples with high values. The variables should be log-transformed prior to the regression analysis to ensure a better contribution of the whole range of values to this relationship.

[Response]

We performed the regression analysis using log-transformed data, but it showed almost similar results. *P*-values, rather, was decreased for the log-transformed data. The similar results are due to small ranges of both the sedDNA concentration and the catch data. We added these results in the Supplementary Figs. 5 and 6.